

# KAM
## Kubernetes Access Manager



**Autorzy**: Vera Goriukhina ⦿ · Marek Fiuk⦿ · Dawid Walkiewicz⦿ · Samuel Żołądz⦿

**Opiekun:** Ireneusz Jóźwiak⦿

### Streszczenie

Obecne narzędzia zarządzania dostępem w klastrach Kubernetes nie oferują wystarczającej elastyczności w definiowaniu uprawnień użytkowników. Opracowane rozwiązanie umożliwia precyzyjne przypisywanie uprawnień w oparciu o przestrzenie nazw (namespace) oraz typy zasobów (resource), zapewniając pełną dowolność konfiguracji. Wykorzystanie standardu OpenID Connect (OIDC) pozwala na integrację z istniejącymi systemami zarządzania tożsamością, takimi jak Keycloak, umożliwiając dalszą ich konfigurację, np. poprzez połączenie z LDAP lub innymi mechanizmami federacji użytkowników.

Rozwiązanie jest dedykowane zespołom, które z uwagi na wymagania bezpieczeństwa potrzebują ograniczać dostęp do klastrów Kubernetes. Kluczowe funkcje obejmują precyzyjne zarządzanie rolami oraz intuicyjną aplikację webową, umożliwiającą wykonywanie operacji zgodnie z nadanymi uprawnieniami. Projekt znacząco zwiększa bezpieczeństwo i efektywność zarządzania infrastrukturą Kubernetes, pozwalając organizacjom na elastyczne dostosowanie uprawnień do indywidualnych potrzeb.

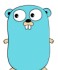 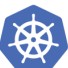 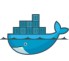 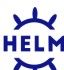 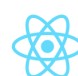 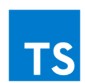 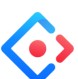 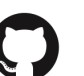 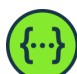 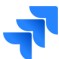

## 1 WSTĘP

Kubernetes to otwarto-źródłowa platforma do zarządzania kontenerami, stworzona przez firmę Google, która obecnie jest rozwijana przez Cloud Native Computing Foundation (CNCF). Umożliwia ona automatyzację wdrażania, skalowania oraz zarządzania aplikacjami kontenerowymi. Dzięki swojej elastyczności i szerokim możliwościom konfiguracji, Kubernetes stał się standardowym rozwiązaniem w środowiskach chmurowych i on-premise. [1] [3] Jednocześnie jednak zarządzanie dostępem do zasobów klastra Kubernetes pozostaje wyzwaniem, szczególnie w większych zespołach.

Głównym problemem, który zidentyfikowano, jest brak narzędzi umożliwiających precyzyjne przypisywanie uprawnień użytkownikom do określonych zasobów Kubernetes. W większych organizacjach jeden klaster może być wykorzystywany przez wiele zespołów, co tworzy wyzwania odpowiedniego zarządzania dostępem. Domyślny mechanizm Role-Based Access Control (RBAC) pozwala na zarządzanie dostępem na poziomie ról, ale nie posiada on możliwości administrowania użytkownikami. Brak dedykowanego interfejsu graficznego dodatkowo komplikuje korzystanie z RBAC z punktu widzenia użytkowników.

Projekt **Kubernetes Access Manager (KAM)** został zaprojektowany, aby rozwiązać te problemy. Główne cele projektu to:

- Opracowanie wieloużytkownikowej aplikacji webowej umożliwiającej przeglądanie i zarządzanie zasobami Kubernetes, takimi jak nody, pody, deploymenty czy config-mapy, z uwzględnieniem szczegółowych uprawnień użytkownika.

- Wprowadzenie systemu zarządzania dostępem opartego na pięciu typach uprawnień: widok listy, widok szczegółowy, edycja, usuwanie i tworzenie zasobów.

- Integracja z zewnętrznym systemem zarządzania użytkownikami pozwaląca na łatwe wdrożenie aplikacji do istniejących organizacji posiadających własnych dostawców tożsamości, takich jak Keycloak.

- Możliwość instalacji na klastrze Kubernetes.

Pod względem praktycznym, aplikacja **KAM** adresuje potrzeby zespołów administracyjnych i deweloperskich, które wymagają precyzyjnego zarządzania dostępem do klastrów Kubernetes. Główne korzyści z wdrożenia systemu to zwiększone bezpieczeństwo operacyjne, łatwość konfiguracji uprawnień oraz uproszczone zarządzanie zasobami w środowiskach wieloosobowych.

## 2   PRACE ZWIĄZANE Z TEMATEM

Na wstępnym etapie zespół przeanalizował istniejące rozwiązania. Wspomniany już mechanizm Role-Based Access Control (RBAC) w Kubernetes dostarcza podstawowych narzędzi do definiowania ról i uprawnień, jednak nie pozwala on na zarządzanie użytkownikami i nie dostarcza powiązanego z uprawnieniami graficznego interfejsu. Istniejące opensource'owe aplikacje webowe, takie jak OpenLens [4] czy Weave GitOps [5], które dostarczają zaawansowanych interfejsów użytkownika do zarządzania klastrami Kubernetes, jednak nie spełniają wymagania zarządzania użytkownikami i ich uprawnieniami. Zespół odrzucił możliwość rozwinięcia tych projektów o mechanizmy autoryzacji i zarządzanie użytkownikami ze względu na braki w dokumentacji oraz złożoność obu projektów.

### 2.1   Unikalne cechy projektu

Projekt **Kubernetes Access Manager (KAM)** wprowadza szereg unikalnych funkcjonalności, które wyróżniają go na tle istniejących rozwiązań:

- **Precyzyjne zarządzanie dostępem**: Aplikacja umożliwia definiowanie dostępu do zasobów na pięciu poziomach (widok listy, widok szczegółowy, tworzenie, edycja i usuwanie). Uprawnienia mogą być przypisywane indywidualnym użytkownikom na podstawie ról, z uwzględnieniem przestrzeni nazw (namespace) oraz typu zasobów (np. Pod, ConfigMap).

- **Integracja z systemami tożsamości**: Dzięki wykorzystaniu standardu OpenID Connect, aplikacja zapewnia integrację z istniejącymi systemami tożsamości, co pozwala na wprowadzenie aplikacji do organizacji posiadających własne rozwiązania w tym zakresie, takie jak Keycloak.

- **Intuicyjny interfejs**: Aplikacja oferuje webowy interfejs, który upraszcza zarządzanie zasobami Kubernetes, umożliwiając zarządzanie zasobami z poziomu przeglądarki z uwzględnieniem uprawnień użytkownika.

Te cechy czynią **KAM** bardziej elastycznym i dostosowanym do wymagań zespołów administracyjnych w porównaniu z innymi dostępnymi narzędziami.

### 2.2   Założenia projektowe i ograniczenia

Projekt zakładał wybór technologii, które są zarówno wydajne, jak i dobrze zintegrowane z Kubernetes:

- Backend został zrealizowany w języku **Go**, co zapewnia wysoką wydajność oraz możliwość wykorzystania istniejących bibliotek do interakcji z Kubernetes API m.in. **client-go**.

- Frontend został zbudowany w oparciu o **React** i **Ant Design**, co umożliwiło stworzenie intuicyjnego interfejsu użytkownika. Technologie te zostały wybrane ze względu na ich popularność na rynku i doświadczenie zespołu.

- Do instalacji aplikacji na klastrze użyto technologii **Helm** i **Docker** będące standardem w środowisku Kubernetes.Pozwalają one na łatwą modyfikację konfiguracji oraz instalację aplikacji.

Podczas realizacji projektu napotkano na pewne wyzwania:

- **Ograniczenia czasowe**: Projekt musiał zostać zrealizowany w określonym terminie, co wymagało priorytetyzacji funkcji w celu dostarczenia MVP (Minimum Viable Product) na czas.

- **Testowanie i skalowalność**: Ograniczony czas wpłynął na możliwość pełnego pokrycia aplikacji testami jednostkowymi i integracyjnymi, co stanowi obszar do dalszego rozwoju.

Wybór podejścia opartego na fazach (od stworzenia MVP do rozwijania dodatkowych funkcjonalności) pozwolił na minimalizację ryzyk związanych z zarządzaniem czasem, jednocześnie umożliwiając dalsze rozwijanie projektu w przyszłości.

## 3   WYNIKI

### 3.1   Funkcjonalności zaimplementowane w aplikacji

Projekt stanowi zaawansowane rozwiązanie umożliwiające zarządzanie zasobami Kubernetes oraz aplikacjami helmowymi za pomocą intuicyjnego panelu webowego (Rysunek 1). Aplikacja została zaprojektowana z myślą o wygodzie użytkownika, oferując szeroką gamę funkcji, takich jak:

- tworzenie i edycja zasobów Kubernetes

- przywracanie aplikacji helmowych do wcześniejszych wersji (rollback)

- usuwanie zasobów oraz aplikacji helmowych

- wyświetlanie listy i szczegółów zasobów Kubernetes oraz aplikacji helmowych

Dzięki temu aplikacja jest odpowiednia zarówno dla mniej zaawansowanych użytkowników, jak i dla administratorów systemów.

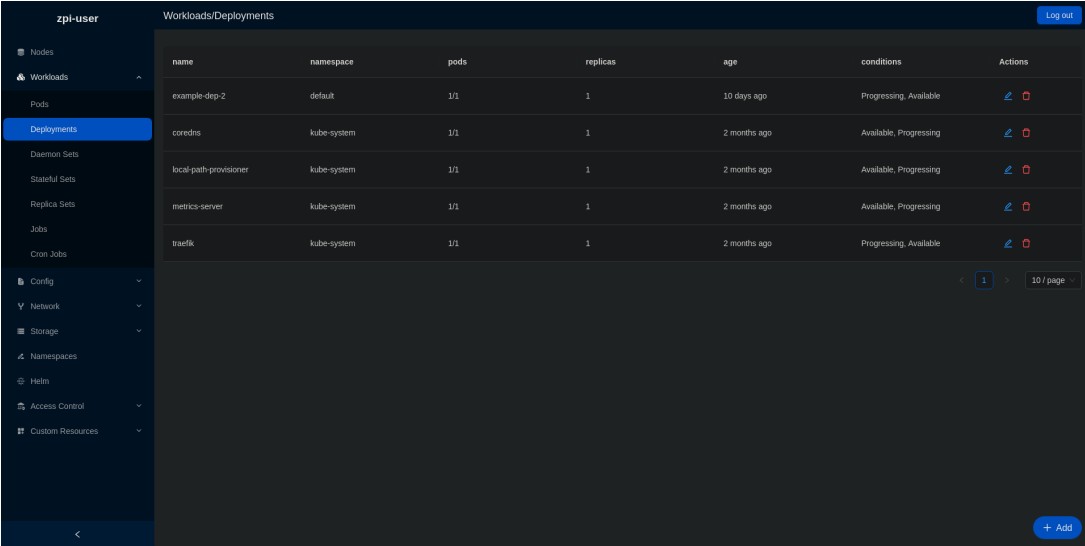

Rysunek 1: Zrzut ekranu z panelu webowego aplikacji.

Kluczową cechą systemu jest elastyczne zarządzanie uprawnieniami, które definiowane są w **ConfigMapie** (Rysunek2). Użytkownicy z wykorzystaniem ról mogą przypisywać uprawnienia w zależności od

```
1   apiVersion: v1
2   data:
3     role-map: |
4       zpi-role:
5         name: "zpi-role"
6         deny:
7           - resource: "Pod"
8             namespace: "*"
9             operations: ["*"]
10        permit:
11          - resource: "*"
12            namespace: "*"
13            operations: ["*"]
14        subroles:
15          - "user"
16      user:
17        name: "user"
18        permit:
19          - resource: "Pod"
20            namespace: "default"
21            operations: ["read"]
22        subroles:
23          - "guest"
24      guest:
25        name: "guest"
26        permit:
27          - resource: "Pod"
28            namespace: "default"
29            operations: ["list"]
30    subrole-map: |
31      superadmin:
32        name: "superadmin"
33        permit:
```

Rysunek 2: Fragment zawartości configmapy przypisującej uprawnienia dla ról.

rodzaju zasobów oraz namespace'ów, co eliminuje ograniczenia typowe dla innych rozwiązań, oferujących jedynie pełny dostęp lub jego brak. Dodatkowo, dzięki wykorzystaniu podról, istnieje możliwość tworzenia szablonów uprawnień, które przyspieszają konfigurację dostępu dla poszczególnych ról.

Aplikacja obsługuje autoryzację i autentykację za pomocą standardu **OIDC** (OpenID Connect), umożliwiając integrację z popularnymi menedżerami dostępu, takimi jak **Keycloak**. Rozwiązanie to pozwala na łatwe podłączenie istniejących federacji użytkowników, delegując proces logowania do zewnętrznego

dostawcy. Taka architektura umożliwia zapewnienie wysokiego poziomu bezpieczeństwa, łatwą aktualizację polityk dostępu oraz integrację z zewnętrznymi systemami bez konieczności modyfikacji kodu aplikacji.

System udostępnia również dobrze udokumentowane **API** (Rysunek3), umożliwiające integrację z innymi aplikacjami oraz automatyzację operacji. Dzięki temu organizacje mogą budować własne narzędzia zarządzania klastrami Kubernetes w oparciu o istniejący system autoryzacji. Dla aplikacji przygotowano

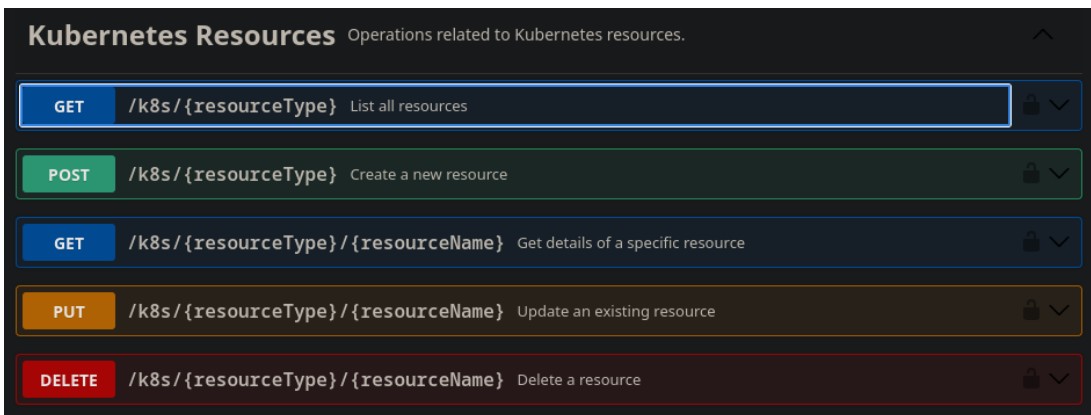

Rysunek 3: Fragment dokumentacji napisany w OpenAPI 3.0 dla enpointów kubernetesowych.

**Helm Chart**, który znacząco upraszcza i przyspiesza jej instalację. Użytkownicy mogą wdrożyć aplikację w środowisku Kubernetes za pomocą jednej komendy, jednocześnie dostosowując konfigurację do specyfiki swojego klastra.

Aplikacja wyróżnia się:

· **Elastycznością** — umożliwia precyzyjne dostosowanie poziomu dostępu do zasobów.

· **Bezpieczeństwem** — wsparcie dla OIDC pozwala na korzystanie ze sprawdzonych mechanizmów autentykacji.

· **Łatwością użytkowania** — intuicyjny interfejs i szybka konfiguracja czynią aplikację przyjazną dla użytkowników o różnym poziomie zaawansowania.

· **Łatwością instalacji** — aplikacja helmowa gwarantuje bezproblemowe wdrożenie i możliwość szybkiej adaptacji w istniejących środowiskach Kubernetes.

## 3.2 Cele techniczne i biznesowe

Głównym celem projektu było zwiększenie elastyczności i bezpieczeństwa w zarządzaniu uprawnieniami w klastrze Kubernetes. Wprowadzenie możliwości definiowania dostępów za pomocą **ConfigMapy** pozwala na szybkie dostosowywanie uprawnień bez potrzeby restartu aplikacji. Wcześniejsze rozwiązania oferowały ograniczoną konfigurację, co prowadziło do nadmiernych uprawnień lub trudności w zarządzaniu.

Nowe podejście znacząco usprawnia proces zarządzania dostępami, zwiększając bezpieczeństwo systemu oraz ograniczając ryzyko błędów, takich jak przypadkowe usunięcie zasobów. Projekt wspiera proaktywne podejście do zarządzania uprawnieniami, co pozytywnie wpływa na efektywność operacyjną organizacji.

## 3.3 Praktyczne zastosowanie

Aplikacja jest obecnie testowana wewnętrznie, jednak jej docelowe zastosowanie obejmuje organizacje, które potrzebują precyzyjnej kontroli dostępu do zasobów. Dzięki przejrzystemu interfejsowi oraz łatwej integracji z istniejącymi federacjami użytkowników, wdrożenie aplikacji przebiega sprawnie i szybko.

Wśród korzyści biznesowych warto wymienić:

· Zmniejszenie ryzyka przypadkowych błędów, takich jak usunięcie krytycznych zasobów.

· Podniesienie poziomu bezpieczeństwa dzięki precyzyjnej kontroli dostępu.

· Zwiększenie efektywności zespołów dzięki uproszczonej obsłudze i zaawansowanej konfiguracji uprawnień.

Rozwiązanie pozwala uniknąć potencjalnych incydentów bezpieczeństwa, minimalizując ryzyko przestojów i strat finansowych, jednocześnie zapewniając stabilność i kontrolę w środowiskach Kubernetes.

## 4    WNIOSKI

Opracowany projekt **Kubernetes Access Manager (KAM)** stanowi odpowiedź na wyzwania związane z zarządzaniem uprawnieniami w klastrach Kubernetes. System umożliwia precyzyjne definiowanie ról i uprawnień na poziomie zasobów i przestrzeni nazw, zapewniając elastyczność konfiguracji oraz przejrzysty interfejs użytkownika.

Kluczowym sukcesem projektu jest integracja z systemami zarządzania tożsamością, takimi jak Keycloak, dzięki czemu aplikacja może zostać szybko wdrożona w istniejących środowiskach organizacyjnych. Dzięki wsparciu dla standardu OpenID Connect, rozwiązanie spełnia wysokie wymagania bezpieczeństwa, jednocześnie oferując łatwą konfigurację.

Dla docelowej grupy odbiorców, w tym zespołów administracyjnych i deweloperskich, aplikacja przynosi znaczące korzyści, takie jak:

- Redukcja ryzyka związanego z nadmiernymi uprawnieniami.

- Ułatwienie zarządzania dostępami w środowiskach wielozespołowych.

- Zwiększenie efektywności dzięki usprawnionemu procesowi konfiguracji.

Projekt **KAM** stanowi fundament dla dalszego rozwoju rozwiązań wspierających zarządzanie klastrami Kubernetes i bezpieczeństwem infrastruktury IT.

## 5    KIERUNKI ROZWOJU

W przyszłości projekt **Kubernetes Access Manager** może zostać rozwinięty o następujące funkcjonalności i usprawnienia:

1. **Zaawansowana analiza logów i audyt operacji**

    - Dodanie modułu monitorującego operacje użytkowników na klastrze i generującego szczegółowe raporty dotyczące aktywności.

    - Możliwość konfiguracji alertów dla operacji o krytycznym charakterze (np. usunięcie zasobu w określonym namespace).

2. **Rozbudowa interfejsu użytkownika**

    - Stworzenie spersonalizowanych widoków dostosowanych do różnych typów zasobów.
    - Wprowadzenie funkcji podglądu historii zmian w konfiguracji uprawnień.
    - Dodanie strony podglądu uprawnień użytkownika.

## 6    PODZIĘKOWANIA

Przede wszystkim chcielibyśmy podziękować prof. dr hab. inż. Ireneuszowi Jóźwiakowi oraz mgr inż. Piotrowi Jóźwiakowi za wsparcie merytoryczne, cenne uwagi oraz inspirację, które odegrały kluczową rolę w realizacji tego projektu.

## LITERATURA

[1] Kubernetes github repository. https://github.com/kubernetes.

[2] Kubernetes access manager project. https://github.com/ZPI-2024-25/KubernetesAccessManager.

[3] Kubernetes project main site. https://kubernetes.io/.

[4] Openlens github repository. https://github.com/MuhammedKalkan/OpenLens.

[5] Weave gitops github repository. https://github.com/weaveworks/weave-gitops.
