# OpenReview forum: "Kubernetes Access Manager"
_pwr.edu.pl/Wrocław_University_of_Science_and_Technology/2024/ZPI_Day — Wrocław University of Science and Technology 2024 ZPI Day Submission_

### Official Review · Reviewer_j1XR · 2024-12-04
**Wartościowa praca wypełniająca brakujące funkcjonalności ekosystemu kubernetes**

**Confidence:** 5
**Significance Of Results:** 4
**Overall Quality:** 4

**Compliance With Template:**

5: Very High Quality – The article contains all the required sections, which are written in a very detailed, clear, and error-free manner. The structure is professional and meets expectations, and the content adheres to the highest substantive and formal standards.

**Description Of Results:**

4: High Quality – The results are described in detail and supported by usage examples or evaluations. The description is reliable but may lack full depth of analysis.

**Feedback On Consistency:**

Artykuł napisany popranie i spójnie językowo. Cel pracy jasno zdefiniowany oraz zrealizowany. Przykłady dobrze dobrane do wymogów dozwolonej długości artykułu.

**Potential For Development:**

Projekt osiągnął stan, w którym dostarcza kompletną średnio rozbudowaną funkcjonalność. Dobór zrealizowanych funkcjonalności dobrze odpowiada wielkości zespołu oraz dostępnego czasu. Projekt posiada spory potencjał na dodatkowe funkcjonalności gwarantujące jego komercyjny sukces.

**Project Nature Evaluation:**

Praca spełnia w całości wymogi stawiane pracom inżynierskim. Przedstawione rozwiązanie jest nowatorskie, dobrze wpisuje się w ekosystem kubernetes. Dostarcza brakujących funkcjonalności nie dostępnych w innych narzędziach tego typu. Dobór technologi wykonany zgodnie ze sztuką. Warto zauważyć, że wybór języka Go bezpośrednio wpisuje się w najczęściej wybierany język dla aplikacji natywnych w kubernetes.
Autorzy wykazali się szczegółową wiedzą dotyczącą systemów autoryzacji oraz autentykacji. Nie starali się implementować istniejących już mechanizmów logowania, korzystając z protokołu OIDC oraz serwisu keycloak.

**Technical Language Precision:**

4: High Quality – The language is appropriate for a technical report. Terminology is used correctly, and statements are precise, with only minor shortcomings that do not affect the overall clarity.

---

### Official Review · Reviewer_zdeV · 2024-12-06
**Kubernetes Access Manager**

**Confidence:** 4
**Significance Of Results:** 4
**Overall Quality:** 4

**Compliance With Template:**

4: High Quality – The article contains all the required sections, which are well-written and substantively correct, although minor errors or shortcomings may be present. The overall structure is clear and coherent.

**Description Of Results:**

4: High Quality – The results are described in detail and supported by usage examples or evaluations. The description is reliable but may lack full depth of analysis.

**Feedback On Consistency:**

Projekt jest poprawnie zorganizowany, zawiera odpowiednie sekcje, z których każda sekcja jasno odnosi się do określonych aspektów projektu. Stosowana terminologia techniczna jest spójna i konsekwentnie używana w całym dokumencie. Styl dokumentu ma techniczny charakter, jest precyzyjny i odpowiedni dla odbiorców zainteresowanych rozwiązaniami z zakresu infrastruktury IT.
Treść wprowadzenia jest spójna z późniejszym opisem wyników i wniosków. Problem zidentyfikowany we wstępie (zarządzanie dostępem w Kubernetes) jest bezpośrednio adresowany przez proponowane rozwiązanie. Cele techniczne i biznesowe są dobrze powiązane z wynikami i funkcjonalnościami projektu, co zwiększa spójność całości.

**Potential For Development:**

Warto wzbogacić literaturę o tzw. zweryfikowane źródła wiedzy. A w przyszłości ujednolić poziom szczegółowości w opisie funkcjonalności aplikacji. Bardzo dobrze byłoby też zadbać o równomierny poziom szczegółowości w każdej sekcji, szczególnie w częściach dotyczących praktycznego zastosowania i analizy wyników.

**Project Nature Evaluation:**

Projekt skutecznie adresuje kluczowe wyzwania związane z zarządzaniem dostępem w klastrach Kubernetes, wprowadza precyzyjne mechanizmy kontroli dostępu oparte na rolach, z integracją przestrzeni nazw i typów zasobów, a dzięki wykorzystaniu standardu OpenID Connect (OIDC) umożliwia bezproblemowe zarządzanie tożsamością. Projekt cechuje się jasno określonym celem, sprecyzowanymi założeniami i dużym potencjałem praktycznym, szczególnie dla organizacji wymagających wysokiego poziomu bezpieczeństwa i efektywności zarządzania zasobami. Bardzo ciekawa praca, wolna od istotnych błędów merytorycznych czy językowych.

**Technical Language Precision:**

4: High Quality – The language is appropriate for a technical report. Terminology is used correctly, and statements are precise, with only minor shortcomings that do not affect the overall clarity.

---

### Official Review · Reviewer_s5y1 · 2024-12-06
**Recenzja projektu Kubernetes Access Manager**

**Confidence:** 4
**Significance Of Results:** 4
**Overall Quality:** 4

**Compliance With Template:**

5: Very High Quality – The article contains all the required sections, which are written in a very detailed, clear, and error-free manner. The structure is professional and meets expectations, and the content adheres to the highest substantive and formal standards.

**Description Of Results:**

4: High Quality – The results are described in detail and supported by usage examples or evaluations. The description is reliable but may lack full depth of analysis.

**Feedback On Consistency:**

Artykuł napisany jest w sposób przejrzysty i spójny, prostym, zrozumiałym jęzkiem. Zawiera rownież wszystkie wymagane elementy, które wprowadzane są w logiczny sposób. Występują jedynie pojedyczne błędy językowe (np. użycie kalki językowej "autentykacja" zamiast polskiego terminu "uwierzytelnienie).
W mojej opinii prezentacja wyników projektu mogłaby być lepsza. Projekt ma charakter konstrukcyjny i dotyczy implementacji rozwiązania odpowiedzialnego za kontrolę uprawnień z klastrze Kubernetes, natomiast na podstawie artykułu trudno stwierdzić jak się ona odbywa. Bazując na przedstawionych zrzutach ekranu można stwierdzić, że jest to rozwinięcie stosowanego w Kubernetesie rozwiązania kontroli dostępu opartej na rolach, jednak poza ogólnymi stwierdzeniami dotyczącymi zwiększenia bezpieczeństwa, większej kontroli uprawnień czy elastyczności trudno zrozumieć, jak projekt realizuje to usprawnienie. Korzystne z tej perspektywy byłoby przedstawienie nawet pojedynczych scenariuszy testowych lub przypadków użycia, które mogą zostać zrealizowane za pomocą opracowanego rozwiązania.
Chciałbym również zwrócić uwagę na możliwość porównania projektu z innymi dostępnymi na rynku, nie tylko darmowymi. Jednym częściej używanych jest Rancher, który myślę, że warto byłoby uwzględnić w analizie obecnego stanu rynku/wiedzy.

**Potential For Development:**

Praca wskazuje dwa dalsze kierunki rozwoju projektu, wyróżniając dla każdego z nich kilka aspektów. Szczególnie wartościowe z mojej perspektywy jest zauważenie konieczności monitorowania stanu klastra i powiadamiania o sytuacjach potencjalnie niepożądanych, np. usunięcie zasobu.
Warto byłoby również dodać tutaj możliwości potencjalnej komercjalizacji projektu, z uwzględnieniem zmian, które należałoby wprowadzić dla poszczególnych grup klientów.

**Project Nature Evaluation:**

Praca ma z pewnościa charakter inżynierski, a wybrane metody techniczne są adekwatne do omawianego problemu. Szczególnie pozytwnie odbieram fakt integracji z KeyCloak, jako przykładowym rozwiązaniem klasy SSO, które w większych przedsiębiorstwach stanowi podstawę zapewnienia kontroli uprawnień użytkowników.
Kontrola dostępu jest jednym z elementów zapewnienia bezpieczeństwa, które jest również często przywoływanym w tekście aspektem zrealizowanego projektu. Z tego względu korzystne byłoby określenie, w jakim zakresie wdrożenie przygotowanego rozwiązania poprawi bezpieczeństwo - czy będzie to zwiększenie poufności, zapewnienie integralności czy może odporność na awarie i możliwość monitorowania stanu zasobów i aktywności użytkowników. Warto byłoby również uzasadnić, w jaki sposób można ocenić wzrost poziomu bezpieczeństwa względem innych rozwiązań.

**Technical Language Precision:**

4: High Quality – The language is appropriate for a technical report. Terminology is used correctly, and statements are precise, with only minor shortcomings that do not affect the overall clarity.

---

### Official Review · Reviewer_tAN3 · 2024-12-06
**The Kubernetes Access Manager (KAM) project has been described as a coherent, logical and practical solution that effectively solves the challenges of managing permissions in Kubernetes clusters.**

**Confidence:** 5
**Significance Of Results:** 4
**Overall Quality:** 4

**Compliance With Template:**

5: Very High Quality – The article contains all the required sections, which are written in a very detailed, clear, and error-free manner. The structure is professional and meets expectations, and the content adheres to the highest substantive and formal standards.

**Description Of Results:**

4: High Quality – The results are described in detail and supported by usage examples or evaluations. The description is reliable but may lack full depth of analysis.

**Feedback On Consistency:**

Yes, the description of the Kubernetes Access Manager (KAM) project is consistent and logical. The problem analysis clearly identifies the challenges of managing permissions in Kubernetes clusters. The presentation of results, such as precise role definition, integration with identity systems and support for OpenID Connect, is well related to the problem presented. The conclusions highlight the practical benefits for users and opportunities for further development, which coherently closes the entire project description.

**Potential For Development:**

The article clearly indicates opportunities for further development and practical application of the Kubernetes Access Manager project results. The directions indicated, such as advanced log analysis and operations auditing or expansion of the user interface, show that the authors have a clear vision for further improvements. Suggestions such as personalized resource views, history of changes in privilege configurations or alerts for critical operations are practical solutions that can increase the usability of the project in real-world scenarios. This demonstrates that the project's results have great potential for further development and practical application.

**Project Nature Evaluation:**

Yes, the Kubernetes Access Manager (KAM) project exhibits features of engineering work. It combines advanced technical methods, such as integration with identity management systems (e.g., Keycloak) and support for OpenID Connect, with high usability for administrative and development teams. The design provides configuration flexibility, security and a clear interface, confirming its practical and engineering approach.

**Technical Language Precision:**

4: High Quality – The language is appropriate for a technical report. Terminology is used correctly, and statements are precise, with only minor shortcomings that do not affect the overall clarity.

---

### Decision · Program_Chairs · 2024-12-10

Accept (Poster)